# The Mechanism of Acupuncture Regulating Autophagy: Progress and Prospect

**DOI:** 10.3390/biom15020263

**Published:** 2025-02-11

**Authors:** Jing He, Min He, Mengmeng Sun, Hongxiu Chen, Zhiqiang Dou, Ru Nie, Jun Zhou, Qingqing Tang, Cong Che, Jie Liu, Tie Li

**Affiliations:** 1Department of Acupuncture and Tuina, Changchun University of Chinese Medicine, Changchun 130117, China; hejing199801@163.com (J.H.); chxxiu525@163.com (H.C.); douzhiqiang12345@163.com (Z.D.); nieru19981229@163.com (R.N.); tqq9723@126.com (Q.T.); 15005525978@163.com (C.C.); 2Northeast Asia Research Institute of Traditional Chinese Medicine, Changchun University of Chinese Medicine, Changchun 130117, China; sunmm@ccucm.edu.cn; 3Research Center of Experimental Acupuncture Science, Tianjin University of Traditional Chinese Medicine, Tianjin 301617, China; axg0820@163.com; 4Academic Affairs Office, Changchun University of Chinese Medicine, Changchun 130117, China; ccucmliujie@163.com

**Keywords:** autophagy, acupuncture therapy, acupoint, mechanism, signaling pathway

## Abstract

Autophagy plays a crucial role in the physiopathological mechanisms of diseases by regulating cellular functions and maintaining cellular homeostasis, which has garnered extensive attention from researchers worldwide. The holistic regulation and bidirectional regulation effects of acupuncture can modulate cellular autophagy, promoting or restoring the homeostasis of the body’s internal environment to achieve therapeutic outcomes. This paper systematically reviews the research progress on the use of acupuncture for treating various diseases via the autophagy pathway, summarizes signal pathways related to acupuncture regulating autophagy, and analyzes the deficiencies present in the existing research. The review results indicate that the mechanism of action of acupuncture on autophagy dysfunction is reflected in the changes in LC3, Beclin1, p53, and autophagy-associated (ATG) protein expression, and regulates signaling pathways and key proteins or genes. The regulatory effect of acupuncture on autophagy capacity is bidirectional: it inhibits the abnormal activation of autophagy to prevent exacerbation of injury and reduce apoptosis, while also activating or enhancing autophagy to promote the elimination of inflammation and reduce oxidative stress. Further analysis suggests that the mechanisms of acupuncture regulating autophagy are insufficiently explored. Future research should prioritize the development of more appropriate animal models, analyzing the accuracy of relevant pathways and the specificity of indicators, exploring the synergistic effects among targets and signaling pathways, clarifying the regulatory mechanisms of acupuncture at various stages of autophagy, and evaluating the efficacy of acupuncture in autophagy modulating. This paper offers valuable insights into the regulation of autophagy by acupuncture.

## 1. Introduction

Autophagy is a complex molecular pathway by which eukaryotic cells degrade long-lived proteins, organelles, and pathogens, and recycle some of the energy required for recirculation to maintain cellular and organismal homeostasis. Autophagy is involved in intercellular communication; responding to cellular stress; regulating cell survival, senescence, and death; inhibiting inflammatory responses; and improving organismal metabolism. Recent studies have demonstrated that the modulation of autophagy targets is an important new pathway for anti-aging, enhancing the effects of targeted cancer therapies, combating viral infections, prolonging lifespan, and treating metabolic disorders, neurodegenerative diseases, and autoimmune diseases [1,2,3,4].

Acupuncture is a natural therapy with a history of more than 3000 years and is used to treat many clinical diseases, signal transduction is one of its key mechanisms. An increasing number of studies have shown that acupuncture causes changes in autophagy-related regulators and signaling pathways during the treatment of neurological, respiratory, and digestive diseases [5,6,7]. This paper summarizes the crosstalk between autophagy and pathophysiological mechanisms, reviews the current findings on the mechanism of action and results of acupuncture in treating various diseases through the autophagy pathway, and discusses the relevant signaling pathways of acupuncture in regulating autophagy to provide new insights into the mechanisms underlying the treatment of diseases with acupuncture, improve the precision and effectiveness of acupuncture therapy, and formulate a more scientific and rational treatment plan.

## 2. Autophagy and Its Influential Role in Pathophysiologic Mechanisms

In mammals, autophagy is categorized into several stages: initiation, phagophore nucleation, phagophore amplification, autophagosome formation, autophagosome-lysosome fusion, and lysosomal substrate degradation [8]. During the initiation phase, double-membrane vesicles are formed, extending outward to create autophagic vesicles. Once formed, phagocytic vesicles detach from the endoplasmic reticulum (ER). Atg9-containing vesicles may serve as seeds for establishing membrane contact sites between the ER and the growing phagophore, thereby nucleating autophagosome biogenesis [9]. Nucleated phagocytic vesicles elongate to form autophagosomes with the participation of two ubiquitin-like protein conjugation systems, ATG12 and ATG8, as well as other related components, and recognize and wrap around autophagic substrates under complex and delicate regulatory mechanisms [10]. Subsequently, the outer membrane of the autophagosome, which encloses the contents, fuses with the lysosomal membrane. This process allows the inner membrane to enter the lysosome, resulting in the formation of an autolysosome [11]. Substances in the autophagic lysosomes are degraded by lysosomes enriched with acid hydrolases. The resulting degradation products, including amino acids, proteins, carbohydrates, and other cellular constituents, are transported to the cytosol to synthesize new cellular constituents or provide energy. The residues are either excreted or retained in the cytosol to maintain homeostasis in the internal environment.

Autophagy is involved in a range of biological processes and is closely linked to numerous pathophysiological mechanisms, including cellular stress, cell proliferation and differentiation, senescence, death, inflammatory responses, and metabolism (Figure 1). Excessive or insufficient autophagy is detrimental to cellular homeostasis, and it is essential to explore treatments for human diseases based on the biological significance of autophagy.

### 2.1. Cellular Stress

#### 2.1.1. Hypoxic Stress

Under hypoxic conditions, mitochondrial adenosine triphosphate (ATP) production is reduced in aerobic organisms, which cannot perform everyday life activities through cellular respiration [12]. To adapt to hypoxia and maintain cellular homeostasis, a shift in energy metabolism occurs, initiating anaerobic glycolysis to obtain energy in an oxygen-independent manner [13]. Hypoxia affects cellular metabolism and is closely associated with pathological conditions such as cardiovascular disease, dementia, and cancer [14]. Autophagy can fine-tune metabolic conditions to induce cellular adaptation to stressful conditions and to maintain cellular homeostasis. Studies have shown that HIF-1α dominates the cellular mechanisms triggered in response to hypoxia [15], and its downstream target gene expression induces the expression of autophagy-related genes and the formation of autophagic vesicles, which promote autophagy and cause cellular damage [16]. In addition, hypoxia can directly regulate autophagy-related signaling pathways and genes, such as AMPK, mTOR, Beclin1, ULK1, and ATG proteins to trigger autophagy [17,18].

#### 2.1.2. Temperature Stress

Biological responses to temperature changes depend on epigenetic regulation [19], which can activate heat/cold stress responses (heat/cold shock response) that affect biological processes such as translation, DNA repair, and DNA synthesis [20]. The heat shock response leads to structural alterations, damage, and protein degradation, thereby affecting cellular functions [21]. The expression of heat shock proteins (HSPs) and heat shock factors plays a key role in the molecular diagnosis and treatment of human diseases, including cancer, aging, infection, and immunity [22]. Heat shock response induces autophagy gene expression. Small HSPs promote autophagic degradation to eliminate denatured proteins, which plays a role in preventing and treating neurodegenerative diseases [23]. In contrast to heat shock, the cellular cold-shock response has been studied less extensively. Studies have shown [24] that hypothermia decreases human physiological activity and activates indicators of autophagy dysfunction and inflammatory and apoptotic signaling pathways, leading to molecular tissue alterations and cell death, negatively affecting human immunity.

#### 2.1.3. Osmotic Stress

Hypertonic or hypotonic environments trigger cellular swelling or contraction, leading to the aggregation of harmful proteins that impair cell survival [25]. Abnormal cellular osmoregulation negatively affects pathophysiological processes, such as brain edema, cataracts, and aging [26]. Autophagy contributes to the elimination of harmful protein aggregates and improves cellular survival. Hyperosmotic stress causes lens epithelial cells (LEC) to exhibit increased apoptosis and decreased viability. Autophagy is abnormally activated, and it has been hypothesized that enhanced autophagy in LEC is a cellular adaptation mechanism to hyperosmotic stress [27]. In model organisms, it has been observed that osmotic stress-induced α-synuclein aggregates and oxidative stress are mediated through ATG7 and an lgg-1-dependent autophagy pathway, which provides a potential link between autophagy dysfunction and prominent proteinopathies, such as Parkinson’s disease (PD) [28]. Hypotonic stress leads to cytoplasmic acidification, which significantly alters the intracellular trophic homeostasis. In the liver, cellular hypotonic stress activates p38^MAPK^ to inhibit autophagy [29].

#### 2.1.4. DNA Damage and Oxidative Stress

Eukaryotic cells are exposed to various endogenous and exogenous stresses, and DNA damage occurs at an order of magnitude of 10^4^–10^5^ per cell per day [30]. This exposure is accompanied by an excessive accumulation of reactive oxygen species (ROS), which induces oxidative stress and negatively affects cellular function and homeostasis. Furthermore, oxidative stress not only impairs cellular integrity and function but can also lead to DNA damage, either directly or indirectly [31]. Autophagy is recognized as a crucial component of the integrated cellular stress response, providing adaptive and protective mechanisms against DNA damage and oxidative stress in cells. It can be adaptive and protective against cellular DNA damage and can act as an energy source during cell cycle arrest and repair mechanisms [32,33]. DNA damage has been shown to activate autophagy through mTORC1 signaling, stimulate the expression of pro-autophagic p53-induced target genes, and enhance the interaction of the p105 subunit of NF-kappaB with autophagy proteins such as Beclin1 [34,35]. Furthermore, oxidative stress-induced autophagy plays a vital role in preventing cellular damage and maintaining homeostasis in vivo [36] ROS can induce the formation and expansion of autophagosomes, initiating autophagy to protect cells from oxidative stress [37]. Additionally, autophagy is believed to be involved in antioxidant functions and DDR/R. Genetic defects in autophagy genes can lead to tumor development, which is associated with ROS accumulation and subsequent DNA damage and organelles [38].

#### 2.1.5. Endoplasmic Reticulum Stress

The ER is crucial for protein folding, translocation, and post-translational modifications, as well as for Ca^2+^ storage and lipid and carbohydrate metabolism, thereby playing an essential role in maintaining cellular homeostasis and function [39]. ER stress is involved in the onset and progression of various diseases including cancer, diabetes, atherosclerosis, obesity, and neurodegenerative diseases [40,41]. The unfolded protein response, autophagy, and mitochondrial crosstalk are thought to underlie the ER stress response. Autophagy contributes to cell survival after ER stress; however, sustained ER stress triggers autophagy-dependent cell death (ADCD) mechanisms [42]. The intrinsic mechanism of PM2.5, which leads to skin aging, inflammation, and even skin cancer, may be related to PM2.5-induced ER stress and autophagy activation. Lycium barbarum polysaccharide protects skin cells from the cytotoxic effects of PM2.5 through the oxidative stress–ER stress–autophagy–apoptosis signaling axis, thereby providing a protective effect on the skin [43]. ER-phagy, a specific form of microautophagy [44], has been extensively studied for its pathophysiological role in various diseases [45].

### 2.2. Cell Proliferation and Differentiation

Cell proliferation and differentiation compensate for cell loss during metabolism and promote wound healing, tissue regeneration, and pathological tissue repair [46]. Several studies have revealed a correlation between the imbalance between cell proliferation and differentiation and the progression of diseases such as tumors, leukemia, and Fanconi anemia [46,47,48]. Autophagy positively regulates cell proliferation in various cell types [49]. In non-small cell lung cancer (NSCLC), autophagy promotes NSCLC cell proliferation and migration by regulating autophagy and ER stress, revealing a role between cell proliferation and autophagy [47]. Similarly, in studies on the Hidradenitis elegans cryptic nematode model, a positive role for autophagy in stem cell proliferation was determined. [50]. On the other hand, autophagy is one of the factors driving tumor growth, potentially accelerating tumor progression and contributing to cancer recurrence as well as the emergence of drug resistance [51]. Therefore, exploring the mechanism by which autophagy affects cell proliferation is beneficial for understanding disease progression and providing new insights for developing therapeutic strategies.

### 2.3. Cellular Senescence

Cellular senescence was first identified in 1960 [52] and is induced by cellular damage, genomic instability, and oxidative stress [53]. It is considered to be a key factor in aging and age-related diseases. Cellular senescence plays a positive regulatory role in physiological processes, such as embryonic development, wound healing, and restoration of homeostasis. Prolonged senescence amplifies the effects of endogenous cellular proliferative arrest, leading to impaired tissue regeneration, metabolic disorders, aging, and a wide range of diseases [54]. Cellular senescence affects other cellular processes, including autophagy. In recent years, the regulation of cellular senescence using autophagy as an entry point has been investigated in mechanistic and therapeutic studies of various diseases. For example, autophagy gene expression declines with age in the brains of patients with Alzheimer’s disease (AD) [55], and autophagy regulators in the skin are beneficial in preventing skin cell aging [56]. However, whether autophagy plays a positive or a negative role in aging remains controversial [57]. Research has demonstrated that autophagy can accelerate senescence via the TASCC pathway. Conversely, selective autophagy actively inhibits cellular senescence by degrading the senescence regulator GATA4. These findings suggest that the role of autophagy in cellular aging is complex, exhibiting either aging or anti-aging effects depending on the specific type of autophagy involved [57].

### 2.4. Cell Death

Cell death plays a crucial role in maintaining biological homeostasis by facilitating tissue remodeling and the removal of abnormal cells. However, the sudden loss of a large number of cells or the accumulation of cell death can exacerbate disease progression [58]. Cell death patterns are categorized into accidental cell death (ACD) and regulated cell death (RCD), with RCD further divided into three types: apoptosis, necroptosis, and pyroptosis. Additionally, ferroptosis is recognized as a distinct form of iron-dependent RCD. In mammals, dysregulated ferroptosis significantly contributes to various pathological processes and conditions, including acute tissue damage, infections, and cancer. Autophagy is predominantly associated with cell death. The involvement of autophagy in cell death can be categorized into ADCD and autophagy-mediated cell death (AMCD). ADCD is significantly influenced by various components of the autophagy pathway. Autophagy flux is elevated during cell death and operates independently of other forms of programmed cell death. ER-phagy and mitophagy are classified as forms of ADCD. In contrast, AMCD positions autophagy as a foundational process that either initiates other modes of cell death, interacts with cell death molecules, or directly leads to apoptosis, necrosis, and ferroptosis [59]. ATG12, an autophagy regulatory protein, promotes apoptosis and exhibits anti-cancer effects [60]. The mechanisms underlying necroptosis and apoptosis partially overlap, with autophagy mediating the transition between these two processes in specific cellular environments. Necrotic apoptosis facilitates the early stages of autophagy while exhibiting an opposing effect in the later stages [61]. Additionally, autophagy regulates the caspase-1-mediated classical pyroptosis pathway through the NLRP3 inflammasome and ROS in the liver [62]. Recently, ferroptosis has been recognized as frequently occurring alongside the overactivation of autophagy and lysosomes [63,64]. Autophagy inhibits lipid peroxidation, maintains cellular homeostasis, and selectively removes damaged or dysfunctional cellular components during ferroptosis.

### 2.5. Inflammatory Response

Inflammation is a highly organized physiological and immune response that occurs in response to stimulatory signals released by pathogens, damaged cells, or allergens [65]. It protects the host from harmful substances, such as viruses and bacteria, and promotes tissue repair and regeneration. Prolonged or aberrantly activated inflammation may be a key factor in disease progression [66]. The role of autophagy in inflammation has been extensively studied, and the two are interrelated in the immune context [67]. Autophagy and autophagy-related pathways are thought to protect against excessive inflammation, and multiple autophagy receptors that bind to nascent autophagosomes can coordinate the removal of inflammatory sources. Autophagy dysfunction contributes to tissue damage and inflammatory pathology [68]. It has been reported that autophagy is involved in driving and regulating inflammatory responses in the lungs and plays an essential role in the development and pathogenesis of many chronic lung diseases, and targeting autophagy shows excellent potential for treating the disease [69]. Increased intestinal permeability in patients with inflammatory bowel disease (IBD) is associated with the aberrant expression of the tight junction (TJ) protein CLDN2, and autophagy reduces epithelial permeability via CLDN2, which protects the epithelial barrier [70].

### 2.6. Metabolism

Metabolism is a series of reactions within an organism’s cells, including catabolism and anabolism, which maximize captured energy and minimize expended energy, respectively. Metabolic processes involve many interrelated cellular pathways designed to maintain cellular and systemic functions and provide the energy required by organisms. When disturbed or disrupted, metabolism is pathogenic and has been linked to the development of diseases such as genetic disorders, cancer, and various metabolic syndromes [71]. Autophagy plays an essential role in the metabolism and turnover of damaged organelles, proteins, and nucleic acids. Macromolecules are metabolized to glucose, free fatty acids, and ATP for recycling through the autophagy pathway [72]. A mutually regulated relationship between lipid metabolism and autophagy has been identified, and ATG genes are involved in metabolic processes [73]. Disruption of lipid metabolism in cancer cells can lead to systemic energy dysregulation. Stearoyl-CoA desaturase 1 (SCD1) is crucial for maintaining cellular lipid homeostasis and is closely linked to autophagy. This interaction serves as a basis for exploring innovative combinatorial anticancer strategies that utilize SCD1 inhibitors alongside autophagy modulators [74].

## 3. Mechanism of Action of Acupuncture in Regulating Autophagy

The role of autophagy in the development and treatment of diseases has been gradually recognized. Acupuncture, an essential component of traditional medicine, has certain advantages in regulating multiple biological processes and pathways. Acupuncture regulates autophagy in three ways: inhibition, activation, and bidirectional regulation, which is mainly achieved by regulating related factors, protein expression, and signaling pathways. However, the same acupoints have different regulatory effects on autophagy in different diseases [75,76]. Therefore, a more comprehensive and rigorous experimental design is required to explore the specific mechanisms underlying acupuncture-induced autophagy regulation.

### 3.1. Acupuncture Inhibits Autophagy

Diseases for which acupuncture achieves therapeutic effects by inhibiting autophagy include neurological, respiratory, circulatory, digestive, and other related diseases, totaling 13 (Appendix A). The related signaling pathways are mainly mTOR-dependent (Figure 2).

#### 3.1.1. Nervous System

Autophagy removes the accumulated cellular waste by degrading and recycling damaged intracellular organelles and abnormal protein aggregates to maintain nervous system homeostasis. Acupuncture is effective in treating neurological disorders in terms of apoptosis, and the modulation of autophagy may be one of its mechanisms of action. Ischemic stroke (IS) is a cerebrovascular disease characterized by impaired blood flow [77]. The blood supply is severely restricted following IS, which induces excitotoxic cell death and necrotic changes. With an increase in ischemic time, there is a gradual imbalance between energy supply and demand in ischemic brain tissues, resulting in different types of brain cell death [78]. Autophagy is a critical process for cell survival following IS. Electroacupuncture (EA), one of the safest and most effective therapeutic methods for treating IS, can produce neurodegenerative effects on IS, and its mechanism of action is related to the inhibition of autophagy. It was found that [79,80] EA can inhibit neuronal autophagy and improve neurological deficits, improving motor dysfunction and the infarct area in cerebral ischemia model rats. The mechanism of action of EA on post-stroke sequelae has also been shown to be related to the inhibition of neuronal autophagy in the rat hippocampus. Acupuncture was applied at GV14, GV26, GV20, and GV24, demonstrating an effect on post-stroke depression in rats, the mechanism of which is related to the activation of the PI3K/Akt/mTOR pathway [81]; EA reduces post-stroke central pain. This may be achieved by decreasing COX-2/β-catenin protein expression, effectively alleviating chronic stress, inhibiting autophagy, and controlling glial cell activation and neuronal cell apoptosis in the injured zone [82]. In addition, the mechanism by which moxibustion reduces brain tissue damage and protects neurons in rats with cerebral ischemia/reperfusion injury (CIRI) is related to the inhibition of autophagy, which is achieved explicitly by inhibiting the expression of the brain injury-sensitive markers S-100β protein and NSE and activating the autophagy signaling pathway PI3K/AKT/mTOR [83].

#### 3.1.2. Respiratory System

Autophagy plays a key role in the normal function of the lung inflammatory response system and the development and pathogenesis of many chronic lung diseases [69,84]. Acupuncture and moxibustion are effective in treating respiratory diseases and have both advantages and potential. Autophagy plays an essential role in the development and function of lymphocytes and is associated with airway inflammatory response in asthma. Silencing of ATG5, a key gene for autophagy, reduced cytokine secretion and airway hyperresponsiveness to varying degrees, suggesting that ATG5 is involved in the pathogenesis of asthma, whereas acupuncture at GV14, BL13, and ST36 in asthmatic mice inhibited ATG5-mediated autophagy, regulated ER stress, and CD4+ T-lymphocyte differentiation, and effectively attenuated asthmatic airway inflammation [85]. The 2021 Global Chronic Obstructive Pulmonary Disease (COPD) Disease Initiative explicitly suggested that acupuncture improves patients’ quality of life [86]. One study found that EA-attenuated airway and lung inflammatory responses in COPD rats were associated with the modulation of AMPK and mTOR expression and the downregulation of autophagy levels in lung tissues [87].

#### 3.1.3. Circulatory System

Circulatory system diseases mainly refer to cardiovascular diseases [88], and pathological changes in cardiomyocytes are an important pathological link in the development of cardiovascular diseases [89]. Prevention of autophagy may be a new therapeutic target for cardiovascular diseases [90]. Chronic heart failure (CHF) is associated with necrosis, apoptosis, and autophagy in cardiomyocytes as well as the overactivation of the neuroendocrine system. Studies have shown that moxibustion has a significant effect on alleviating heart failure, and its mechanism was found to be related to lowering serum NT-proBNP levels and downregulating p53 to upregulate mTOR expression and inhibit cardiomyocyte autophagy in CHF model rats [91]. The pathogenesis of myocardial ischemia-reperfusion injury involves oxidative stress, ER stress, and autophagy. EA preconditioning has a significant protective effect on ischemia/reperfusion (IR)-injured myocardium, and the mechanism may be related to the inhibition of myocardial tissue autophagy and iron death through the activation of the mTOR/ROS signaling pathway [64].

#### 3.1.4. Digestive System

Autophagy is key to maintaining intestinal homeostasis, regulating the interactions between the gut microbiota and innate and adaptive immunity, and host defense against intestinal pathogens [70]. Acupuncture is effective in improving gastrointestinal function and treating related diseases. Modulation of autophagy abnormalities may be a potential mechanism of action for acupuncture in the treatment of gastrointestinal diseases. The primary pathophysiological mechanism of functional dyspepsia (FD) is related to gastrointestinal dyskinesia. Cajal interstitial cells (ICC) are pacemaker cells in the gastrointestinal tract that regulate intestinal peristalsis; their absence leads to gastrointestinal dyskinesia. EA at ST36 can improve gastrointestinal dyskinesia in FD rats, and the mechanism is related to maintaining the number and structure of ICC and inhibiting excessive autophagy of ICC in FD rats [76]. Crohn’s disease (CD) is a complex, chronic, and nonspecific IBD, and its pathogenesis is closely associated with autophagy-related genetic susceptibility [92]. Abnormalities in autophagy disrupt its interactions with inflammatory cytokines, leading to severe intestinal inflammatory responses and IBD [64]. A study explored the mechanism of action of moxibustion in Crohn’s from the perspectives of autophagy and immunity and found that autophagic activity was reduced in the colonic tissues of CD rats after moxibustion treatment, and the levels of inflammatory factors in the colon of CD rats were significantly downregulated [93].

#### 3.1.5. Other

The mechanism of action of acupuncture in other diseases is also associated with the inhibition of autophagy. The pathogenesis of vascular dementia, an advanced neurocognitive dysfunction syndrome caused by brain injury, is associated with autophagy in hippocampal neurons and a chronic inflammatory response due to chronic cerebral underperfusion [94]. EA pretreatment at GV20, GV14, and BL23 alleviated learning memory impairment, improved neuronal ultrastructure in the hippocampal CA1 area, and repaired damaged neurons in rats with vascular dementia. This mechanism may be related to a reduction in excessive autophagy in neurons, inhibition of the activation of NLRP3 inflammatory vesicles, and attenuation of the central inflammatory response [95]. In addition, EA inhibits neuronal cell autophagy, reduces neuronal cell apoptosis and inflammation in postherpetic neuralgia rats, and improves mechanical pain threshold [96]; participates in the regulation of autophagy-related proteins, promotes white fat browning, and ameliorates obesity [97]; activates the key signaling pathway of autophagy, PI3K/mTOR, and regulates autophagy-related factor (Beclin1, LC3, and p62) expression, inhibits autophagy, and improves facial nerve function [98].

Recent research has increasingly focused on the interconnection between autophagy overactivation and various diseases, prompting a series of experimental studies that investigate the efficacy of acupuncture as a treatment modality based on this concept (Figure 3). Acupuncture has been demonstrated to regulate the levels of key autophagy regulators, such as LC3, Beclin1, and ATG, while also inhibiting autophagy through associated signaling pathways across a range of diseases, thereby contributing to disease improvement and treatment outcomes. Among neurological disorders, IS and its sequelae are mentioned most frequently, with acupoints focused on the Governor’s pulse, and GV14 and GV20 being employed most often. The predominant signaling pathways implicated in these processes include PI3K/AKT/mTOR. Notably, the specific acupoint ST36 has been selected for nearly all diseases treated with acupuncture aimed at inhibiting autophagy. This suggests that acupuncture at ST36 may exert inhibitory effects on autophagy through multiple pathways, including the regulation of autophagy-related factors across the nervous, respiratory, and digestive systems, as well as other tissues and organs.

### 3.2. Activation of Autophagy by Acupuncture

In contrast, acupuncture activates autophagy to exert therapeutic effects in 18 diseases, increasing the number of diseases, including tumors, musculoskeletal system diseases, mental disorders, endocrine diseases, and metabolic diseases. In contrast, for neurological and digestive disorders, the diseases, although different, are aimed at the protection of neurological function and reduction in inflammation as the ultimate goal (Appendix A), and summarizing the relevant signaling pathways of acupuncture activation of autophagy (Figure 4).

#### 3.2.1. Nervous System

Acupuncture interventions enhance autophagy in brain tissue, exert neuroprotective effects, attenuate brain damage, improve neurological function, and promote recovery from intracerebral hemorrhage [99,100,101,102]. PD is associated with abnormal autophagy in neurons [103]. Studies have confirmed that EA promotes autophagy initiation, autophagic flux, and mitophagy in neurons in four regions of the brain (substantia nigra, striatum, hippocampus, and cortex) of the mice models of PD and improves disease-related motor symptoms [103]. Mitophagy is the selective degradation of mitochondria through autophagy, and melatonin-mediated mitophagy is one of the pathways that inhibit NLRP3 inflammasome activation. The key pathological modifier of post-stroke cognitive impairment (PSCI) is the NLRP3 inflammasome, and EA has been demonstrated to inhibit NLRP3 inflammasome activation and attenuate the inflammatory response by promoting the expression of mitochondrial autophagy-associated proteins (PINK1/Parkin), which exert a multiprotective effect in PSCI [104]. The mechanisms of central nervous system diseases are related to blood–brain barrier (BBB) permeability. EA may regulate the expression of TJ proteins in endothelial cells by affecting integrin binding, autophagy pathways, and calcium signaling pathways, which in turn mediate BBB opening. A study based on transcriptomic technology analysis found that EA activates autophagy to improve BBB permeability by inhibiting TORC2 signaling and significantly downregulating the expression of the related gene Prr5l [105].

#### 3.2.2. Neurocognitive Disorders

Autophagy plays a crucial role in neurodevelopmental processes. Targeted drugs that promote autophagy (e.g, rapamycin) upregulate autophagy and contribute to symptom improvement in neurocognitive disorders [106]. Acupuncture reduces Aβ deposition in brain tissue and improves hippocampal neuronal cell apoptosis; its mechanism of action is closely related to autophagy. EA activates autophagy and reduces neuronal cell apoptosis in the hippocampus of APP/PS1 mice [107]. Moxibustion down-regulates its downstream target, p70S6K, through the mTOR signaling pathway, which inhibits the mTOR1 signaling pathway and enhances autophagy [108], thereby enhancing Aβ clearance and improving cognitive deficits. It has also been shown [109,110,111] that acupuncture and moxibustion regulate autophagy in AD and abnormal Aβ deposition in the body about the PI3K/AKT/mTOR signaling pathway. Genetic and epidemiological evidence suggests that dysfunction of the autophagy-lysosomal pathway (ALP) pathway may accelerate Aβ production, one of the earliest pathological changes of AD. The transcription factor EB (TFEB) is the most important regulator of the ALP pathway, and EA activates the TFEB-mediated ALP to promote lysosomal function, enhances autophagy, and synergistically supports degradation of the NLRP3 inflammatory vesicle component, effectively improving AD mice cognitive ability and neuroinflammation, and inhibiting the progression of Aβ pathology [112]. It has also been found that the mechanism of EA-induced TFEB activation is mediated through multiple upstream kinases (including MTORC1, MAPK1, AKT, and AMPK) [113,114]. Perioperative neurocognitive disorder (PND) is a common perioperative complication in elderly patients and belongs to the category of neurocognitive disorders. Dysfunctional autophagy is closely associated with its occurrence and development. Some studies have illustrated that EA treatment not only effectively alleviated the spatial memory deficits in PND model rats but also inhibited the levels of ROS and IL-1β in the hippocampus of PND rats, significantly increasing the autophagy/mitochondrial autophagy markers PINK1, Parkin, LC3, and Beclin1 in hippocampus expression levels, and the mitochondrial autophagy inhibitor Mdivi-1 inhibited these effects of acupuncture, suggesting that the potential mechanism by which acupuncture reduces the incidence of PND is related to its ability to activate PINK1/Parkin-mediated mitochondrial autophagy to inhibit the expression of ROS and IL-1β [115].

#### 3.2.3. Endocrine or Metabolic Diseases

Autophagy is involved in protein renewal, organelle quality control, cellular metabolism, and innate and adaptive immunity. Polycystic ovary syndrome (PCOS) is a common endocrine-metabolic disorder in which abnormal follicular development is an essential pathological feature. Autophagy in ovarian granulosa cells is a key factor in the manifestation of this pathology. Acupuncture effectively improves sex hormone levels and ovarian tissue autophagy in rats with PCOS, thereby treating ovulation disorders [116,117]. Autophagy is also closely associated with oxidative stress and metabolic processes, and the AMPK/mTOR pathway is an essential autophagic antioxidant pathway. Hyperlipidemia is a disease in which lipid metabolism is abnormal and lipid levels are higher than usual in animals. This severely affects hepatic lipid deformation and autophagy, leading to the inactivation of proteins and transcription factors related to regulating autophagy in hepatocytes, causing liver injury. As a safe and effective complementary therapy, moxibustion can regulate hepatic energy metabolism, activate the AMPK/mTOR pathway, promote hepatocyte autophagy, and significantly reduce serum lipoproteins and hepatocyte injury in hyperlipidemic rats [5].

#### 3.2.4. Digestive System

Autophagy plays a key role in maintaining the stability of the internal intestinal environment, protecting the integrity of the colonic mucosal barrier, and balancing the intestinal microbiota and immune response. Abnormal autophagy can cause gastrointestinal dyskinesia leading to diabetic gastroparesis (DGP), functional constipation (FC), and other related diseases. Experimental studies have confirmed that sinus ICC autophagic flux is significantly impaired in DGP model rats. EA can improve gastric motility by regulating ICC autophagic activity and maintaining normal contraction of gastric smooth muscles [118]. Acupuncture can significantly improve FC symptoms, and not only is its short-term efficacy not inferior to the guideline-recommended first-line drugs, but the advantage of maintaining efficacy for a more extended period after stopping the treatment has been confirmed in a series of high-quality clinical randomized controlled studies. Acupuncture in the treatment of FC promotes intestinal motility, regulates intestinal flora, modulates the brain–gut axis, attenuates intestinal inflammatory responses, and improves rectal hypersensitivity [119]. Gut regulation relies on the bidirectional communication between enteric glial cells (EGCs) and neurons. When an inflammatory response occurs in the gut, EGCs maintain a dynamic balance of immunity through autophagy. It was found that EA intervention promotes the expression of autophagy proteins in EGCs in the colon tissues of FC mice through the PI3K/AKT/mTOR signaling pathway, and this effect can be inhibited by the autophagy inhibitor 3-MA [120]. Ulcerative colitis (UC) is a chronic, non-specific inflammatory disease of the colon and rectum of unknown etiology. Experimental studies have found that decreased autophagy levels in UC model rats are associated with immune dysregulation and increased expression of pro-inflammatory factors and that EA intervention inhibits intestinal inflammatory responses and activates the AMPK/mTOR pathway to promote autophagy in colon cells and improves related symptoms in UC rats [121].

#### 3.2.5. Musculoskeletal System

Some studies have shown that the abnormal activation of mitochondrial autophagy is a key pathological mechanism of muscle atrophy. Acupuncture is indicated for treating musculoskeletal disorders by improving symptoms of pain and dysfunction. EA at ST36 can improve mitochondrial autophagy by activating the AMPK/ULK1 pathway and has a favorable therapeutic effect on exercise fatigue in rats with splenic deficiency syndrome [75]. Rheumatoid arthritis (RA) is a highly heterogeneous, chronic, systemic autoimmune disease characterized by chronic excessive proliferation of synoviocytes. Autophagy is essential for regulating the abnormal proliferation of synovial cells. Some studies have shown that moxibustion can correct the local inflammatory response of the joints, alleviate related symptoms, and repair synovial cell damage in RA. It promotes autophagy in synovial tissue [7,122,123,124]. In addition, snap-needle acupuncture can increase autophagy, improve the inflammatory response, and reduce joint swelling and the arthritis index in rat models [125]. Neurogenic cervical spondylosis (CSR) is characterized by neuropathic pain in the innervated area of the corresponding ganglion. Recent studies have found that apoptosis and necrosis of nerve cells can cause neuropathic pain in the CSR and that autophagy dysfunction plays a key role in this process. Moxibustion has a strong analgesic effect on CRS. Experimental studies have found that moxibustion upregulates the levels of autophagy-related regulators (LC3, LC3II/I, Atg7, Beclin1, and Bcl-2) in the spinal cord tissues of rats in the CRS model, which enhances autophagy and facilitates neuronal repair through the Act A/Smad and Beclin1/GRP78 signaling pathways [126,127]. Knee osteoarthritis (KOA) is a degenerative disease of articular cartilage tissues, whose incidence is positively correlated with aging, and whose pathology is key to articular chondrocyte degeneration and apoptosis. Acupuncture knife treatment upregulated autophagy in knee chondrocytes in KOA rabbits, alleviated oxidative stress, and inhibited chondrocyte necrosis [128]. Some studies have found that needle knives activate PINK1/Parkin signaling pathway-mediated mitochondrial autophagy, improve mitochondrial function, and reduce ROS accumulation to reduce cartilage damage [129]. In addition, the mechanism by which EA attenuates the degree of joint inflammation and chondrocyte morphological damage in SD rats, a model of knee osteoarthritis, is associated with the promotion of chondrocyte autophagy and inhibition of the expression of the NLRP3 inflammasome and MMP-13 protein [130]. Intervertebral disk degeneration (IDD) is associated with disk cell apoptosis and excessive autophagy. It has been demonstrated that moxibustion activates autophagy through HIF-1α/VEGF and reduces apoptosis as its potential mechanism of action in treating IDD [131].

#### 3.2.6. Tumors

The dual role of autophagy in cancer progression and suppression has been increasingly recognized. The dysregulation of autophagy indirectly promotes tumor growth and progression through inflammation and its effects on tumor cells. Colorectal cancer (CRC) is the third most common malignant tumor worldwide and has the second-highest mortality rate [132]. CRC can be treated with surgery, radiotherapy, and targeted therapy. However, patients often develop drug resistance, and there is an urgent need to explore new therapeutic strategies to improve the survival rate of patients with CRC. Cancer occurrence is positively correlated with the severity of the inflammatory response, and the inhibition of the colonic inflammatory response is an effective strategy for slowing colonic carcinogenesis. EA intervention can downregulate the levels of inflammatory factors such as IL-6 and IL-17 in CRC model mice and activate autophagy-related miRNAs through the SIRT1 signaling pathway to promote autophagy connexin expression in colorectal tissues, thus blocking inflammatory-cancerous transformation and retarding tumor cell growth [133,134].

#### 3.2.7. Other

In addition, autophagy can attenuate oxidative stress damage induced by free radicals and ROS and slow down aging. EA can promote the occurrence of hepatic autophagy in aging mice by regulating the autophagy-related signaling pathway AMPK/mTOR/ULK1, which can then inhibit excessive oxidative stress and slow down the process of aging to a certain extent [135]. Experimental studies have found that acupoint burrowing improves age-related changes in appearance, muscle function, and spatial memory loss in aged rats, which are related to the improvement of mitochondrial dysfunction and the regulation of hippocampal autophagy activity via the PINK1/Parkin and PINK1 pathways [136].

The second modulatory effect of acupuncture on autophagy is its capacity to activate or enhance this process. Numerous studies have demonstrated that acupuncture’s role in activating autophagy spans a broad range of diseases, with a more extensive examination of the selection of acupuncture points and specific mechanisms compared to those related to the inhibition of autophagy. Among the diseases most frequently studied are AD, RA, neurogenic cervical spondylosis, and osteoarthritis of the knee (Figure 5). The selection of acupoints for treating the same disease tends to be more consistent; for instance, GV20 is commonly used for AD, while ST36 is often employed for RA. The signaling pathways associated with the activation of autophagy by acupuncture include not only the mTOR pathway but also SIRT1, PINK1, and PINK1/Parkin, among others. Additionally, it has been found that varying durations of moxibustion applied to GV14, while all capable of enhancing autophagy and reducing apoptosis in the spinal cord nerve root tissue cells of CSR rats involve distinct molecular mechanisms in the regulation of these processes. This suggests that both acupoints and intervention modalities may serve as valuable entry points for studying the regulation of autophagy through acupuncture. However, the majority of findings concerning the mechanisms of autophagy activation by acupuncture remain fragmented and require further validation through rigorous experimental design.

### 3.3. Bidirectional Regulation of Autophagy by Acupuncture

Autophagy, whether deficient or excessive, disrupts intracellular environmental homeostasis and plays a dual role in cytoprotection and cell death. The mechanisms by which autophagy contributes to the onset and progression of diseases remain a topic of considerable interest. Excessive autophagy can lead to cell dysfunction and cell death, contributing to the development of various diseases as the excessive accumulation of autophagosomes in cardiomyocytes contributes to heart failure [137], while a similar accumulation in the neuronal cytoplasm exacerbates the symptoms associated with neurological deficits in cases of ischemic brain injury [79]. Conversely, autophagy deficiency hinders the degradation and digestion of long-lived proteins and damaged organelles, which contributes to the development of several diseases, with AD being the most representative. Furthermore, autophagy deficiencies exacerbate amyloidosis and tau pathology [110]. Interestingly, the development of certain diseases is inextricably linked to both overactivation and underactivation of autophagy. For example, the role of autophagy in traumatic brain injury (TBI) remains controversial [138,139]. A review of the relevant literature suggests that the modulation of autophagy through acupuncture can have opposing effects within the same disease context, as observed in conditions such as cerebral ischemia/reperfusion, myocardial ischemia, and TBI (Appendix A).

EA regulates autophagy in the context of cerebral ischemia-reperfusion injury (CIRI), resulting in two primary outcomes. First, it modulates apoptosis following CIRI by activating autophagy. By stimulating the PI3K/Akt and type III PI3K/Beclin1 signaling pathways, EA enhances autophagy, reduces the number of apoptotic cells, and decreases infarct volume, thereby exerting a neuroprotective effect [140,141]. Second, EA inhibits excessive neuronal autophagy by reducing the number of autophagosomes and autophagolysosomes, regulating the expression levels of autophagy-related factors and microRNAs, and mitigating brain injury along with the associated inflammation [6,124,142,143,144,145]. In contrast, only moxibustion pretreatment has been reported to improve CIRI, and its mechanism is associated with autophagy inhibition [83]. The acupoints selected for EA and moxibustion treatment of CIRI predominantly include the Governor’s pulse, with GV14 and GV20 being the most frequently targeted. Autophagy has been shown to protect cardiomyocytes during the early stages of acute myocardial ischemia (AMI). However, in the later stages, excessive autophagic activity may exacerbate myocardial necrosis and dysfunction. The regulation of cardiomyocyte autophagy by acupuncture is also bidirectional. EA intervention at HT5 and HT7 in the heart meridian has demonstrated therapeutic effects by activating the AKT/mTOR signaling pathway and inhibiting autophagy levels in cardiomyocytes of AMI rats [146]. Conversely, the effects of PC6, which interacts with the pericardium meridian, in reducing apoptosis and necrosis while promoting the survival of damaged cardiomyocytes, are linked to the activation of autophagy through the LKB1/AMPK/PFK2 signaling pathway [147].

Apoptosis and autophagy activities in the SCI lesion area were inhibited, which limited nerve regeneration and functional recovery [148]. EA at the EX-B2 acupoints inversely activated autophagy and regulated the disordered microenvironment of the SCI, thereby promoting the recovery of motor function following SCI [149]. Furthermore, EA at the GV4 and GV14 acupoints inhibited autophagy by enhancing the activity of the PI3K/AKT/mTOR signaling pathway, which facilitated the recovery of hindlimb motor function and spinal cord neurological function in rats with SCI. This effect may be associated with the regulation of cell proliferation and differentiation, as well as the promotion of axon regeneration [150]. In post-TBI, acupuncture can either promote or inhibit autophagy, leading to improvements in neurological function, cognitive abilities, and behavioral outcomes in animal models of TBI. EA facilitates the recovery of neurological function and mitigates pathological brain tissue damage and neuronal apoptosis in TBI rats by inhibiting autophagy [139,151]. However, acupuncture exerts a time-dependent effect on neuronal autophagy in the damaged cerebral cortex following TBI. This is evidenced by observations that acupuncture enhances autophagic flux and induces autophagic activation on day 3 post-injury while preventing excessive autophagy on days 7 and 14 post-injury [152]. Thus, acupuncture appears to enhance autophagy and improve TBI prognosis when autophagy levels are insufficient while inhibiting autophagy to exert therapeutic effects when autophagy is excessive. Collectively, these findings indicate that the role of autophagy in disease progression can vary at different pathological stages and that the modulation of autophagy by acupuncture may be closely linked to the timing of the intervention.

As illustrated in Figure 6, acupuncture treatment for CIRI, SCI, and TBI yielded significantly different outcomes in the regulation of autophagy. Autophagy serves distinct roles at various stages of the disease. Zhao et al. [152] demonstrated that the neuroprotective effects of acupuncture on TBI rats are mediated through an autophagy-related mechanism, exhibiting a bidirectional regulatory effect. Notably, on day 3 post-TBI, acupuncture was found to promote neuronal autophagy; conversely, on days 7 and 14 post-TBI, it inhibited neuronal autophagy, with both effects ultimately contributing to disease recovery. This observation may further reflect the bidirectional regulatory capacity of acupuncture. Acupuncture facilitates self-regulation through a homeostatic mechanism [153], allowing it to correct abnormal autophagy—whether excessively activated or functionally impaired—within disease states, thereby restoring body function from a biased state to a balanced one. Consequently, the direction of acupuncture’s effects is contingent upon the pathological condition of the organism.

## 4. Conclusions and Outlook

The human body is a complex system that maintains homeostasis through various homeostatic mechanisms [154]. Acupuncture offers multiple pathways, targets, and effects, demonstrating significant potential for anti-inflammatory, anti-apoptotic, and anti-oxidative stress responses. Additionally, it promotes nerve repair, enhances neurotrophic factors, and modulates the immune response by targeting the autophagy pathway. Acupuncture as an effective physical stimulus, can act as a “promoter” to activate or enhance autophagy, thereby accelerating the removal of pathological products (e.g., Abeta deposited), reducing oxidative stress, improving metabolism, and facilitating tissue repair. Furthermore, in instances where autophagy becomes excessive in organs or tissues due to certain disease states, acupuncture assumes a negative regulatory role, inhibiting autophagy to mitigate apoptosis, decrease inflammatory responses, and prevent excessive tissue damage. According to traditional Chinese medicine, the onset of disease is associated with impaired “qi” function within the meridian system, which may represent an active substance involved in disease treatment. Acupuncture can stimulate specific acupoints on the body surface to regulate “qi” and direct it toward the affected area, thereby producing therapeutic effects [155]. Acupuncture promotes self-regulation through an autostable mechanism, with its direction of action contingent on the body’s pathological conditions. Whether addressing abnormal autophagy resulting from various diseases (either over-activation or under-activation) or autophagy disorders that manifest at different stages of the same disease, acupuncture can help restore the body’s functions from a biased state to a balanced one. This underscores the bidirectional regulatory effect of acupuncture.

Acupuncture stimulation on acupoints enhances the body’s self-regulatory capacity. However, several factors can significantly influence the efficacy and outcomes of acupuncture in regulating autophagy. These factors include the intervention method, timing of treatment, selected meridian acupoints, the nature of the disease, and the individual’s physiological status. Currently, the understanding of the mechanisms by which acupuncture regulates autophagy remains fragmented and preliminary. The existing literature exhibits several shortcomings: (1) a predominant focus on isolated signaling molecules or targets; (2) a reliance on animal models induced by pharmacological agents or chemical reagents to create pathological conditions that do not accurately represent specific traditional Chinese medicine syndrome types or the severity of reactions associated with the disease; and (3) a lack of foundational studies on acupuncture’s regulation of autophagy that have been translated into clinical applications. In summary, these issues have not been thoroughly investigated, and numerous hypotheses remain untested. Consequently, future research on acupuncture’s role in regulating autophagy should prioritize refining the design of foundational studies, developing more appropriate animal models, clarifying the precision of relevant pathways and indicators, exploring the interactions among various targets and signaling pathways, and assessing the effectiveness of acupuncture in modulating autophagy.

With the increasing recognition of autophagy and the expansion of related research, the intricate relationship between autophagy dysfunction and various human diseases has become increasingly evident. Numerous studies have demonstrated that autophagy dysfunction is implicated in the onset and progression of a wide range of diseases, characterized by either abnormal activation or defective autophagy. Research into the modulation of autophagy through acupuncture has indicated that moxibustion applied at different times on the same acupuncture point can enhance or restore autophagy levels in the hippocampus via the PI3K/AKT/mTOR signaling pathway, thereby facilitating the clearance of abnormal Abeta deposited in the body [110,111]. Nevertheless, due to variations in experimental conditions and the selection of detection indicators, a quantitative characterization of the relationship between different durations of moxibustion and their effects on autophagy regulation remains elusive. Furthermore, studies indicate that different acupuncture sessions or moxibustion administered at varying temperatures yield distinct effects on alleviating visceral hypersensitivity reactions and manifest differing analgesic properties [156]. The therapeutic effects of various acupuncture modalities on diseases differ significantly [157]. Autophagy plays a dual role in both promoting and inhibiting diseases, and the regulatory influence of acupuncture on autophagy is not fixed. Identical acupuncture modalities, when applied to the same acupoints and engaging the same signaling pathways, can yield completely opposite effects on autophagy regulation [75,76]. This phenomenon is associated with the holistic and bidirectional regulatory nature of acupuncture. For a given disease, the same intervention modality and selection of acupuncture points can lead to similar autophagy regulation through distinct signaling pathways [101,102]. In conclusion, further analysis of diverse acupuncture techniques and intervention durations is essential for understanding their differential impact on autophagic regulation. It is also crucial to investigate the efficacy thresholds and bidirectional regulatory effects of acupuncture, as well as to elucidate the mechanisms through which acupuncture modulates autophagy at various stages.

Based on the preceding discussion, it is evident that regulating autophagy is one of the mechanisms through which acupuncture exerts its therapeutic effects. This paper summarizes recent advancements in research concerning the regulation of autophagy by acupuncture, addresses the limitations of existing studies, and proposes potential directions for future research in this area. With the ongoing development of contemporary science, technology, and research methodologies, exploring the biological effects of acupuncture on overall regulation will play a significant role in advancing acupuncture medicine and enhancing human health.

## Figures and Tables

**Figure 1 biomolecules-15-00263-f001:**
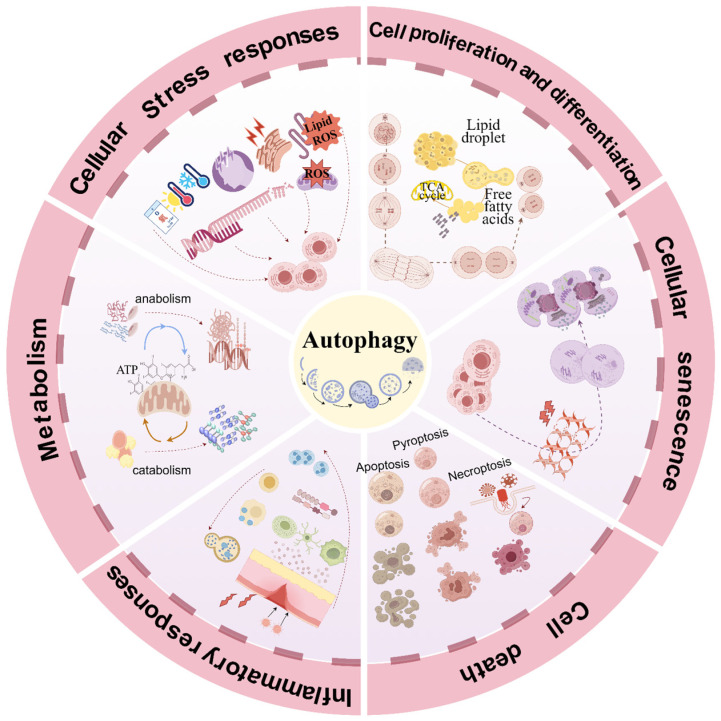
2.0Crosstalk between autophagy and pathophysiologic mechanisms. Autophagy is an essential mechanism in the regulation of cellular homeostasis. Involved in cellular stress, cell proliferation and differentiation, senescence, death, inflammatory response, and metabolism. ROS, reactive oxygen species; Lipid ROS, lipid reactive oxygen species; TCA cycle, tricarboxylic acid cycle; ATP, adenosine triphosphate. By Figdraw 2.0.

**Figure 2 biomolecules-15-00263-f002:**
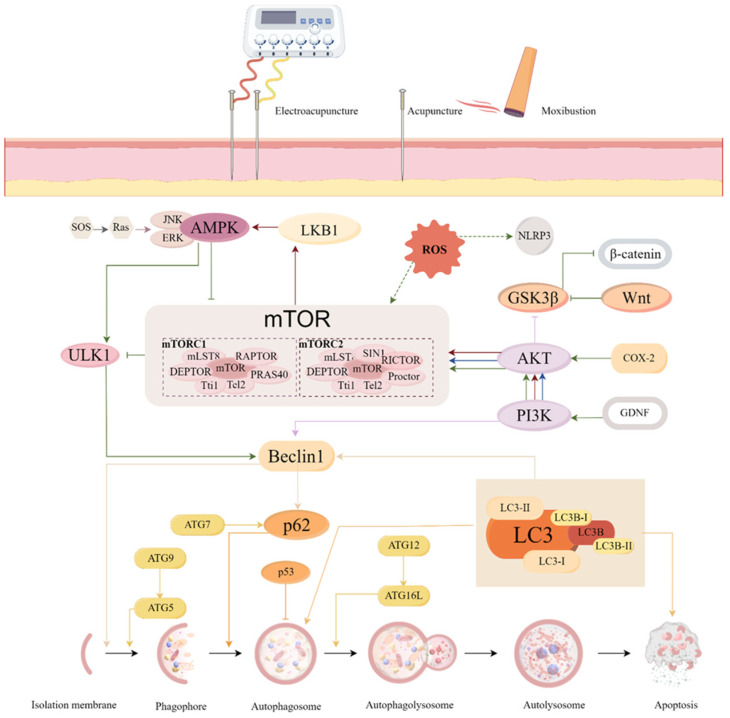
Signaling pathways associated with acupuncture inhibit autophagy. There is increasing evidence that excessive autophagy can lead to cell death, which subsequently causes or exacerbates tissue damage. This phenomenon is strongly associated with the onset and progression of several diseases, including ischemic stroke (IS), asthma, and functional dyspepsia (FD). Acupuncture regulates the homeostasis of the intracellular environment by inhibiting excessive autophagy through relevant signaling pathways, which helps to improve disease symptoms and reduce tissue damage. The different colors of the lines represent different acupuncture therapies: moxibustion: red; acupuncture: blue; electroacupuncture (EA): green. For example, EA inhibited neuronal autophagy through the mTORC1/ULK1/Beclin1 signaling pathway in a rat model of cerebral ischemia to ameliorate neurological deficits. SOS, Son-of-Sevenless; Ras, Ras protein; JNK, c-Jun N-terminal kinase; ERK, extracellular signal-regulated kinase; AMPK, AMP-activated protein kinase; LKB1, liver kinase B1; ULK1, Unc-51 Like Kinase 1; mTOR, mammalian target of rapamycin; ROS, reactive oxygen species; NLRP3, NOD-like receptor thermal protein domain associated protein 3; GSK3β, glycogen synthase kinase-3β; β-catenin, Beta-catenin; Wnt, Wingless-Type MMTV Integration Site Family; COX-2, Cyclooxygenase-2; GDNF, Glial Cell Line-Derived Neurotrophic Factor; AKT, threonine kinase; PI3K, phosphoinositide3-kinase; p62, Ubiquitin-binding protein p62; ATG, Anti-human T-lymphocyte Globulin; LC3, microtubule-associated protein 1 light chain 3; LC3B, microtubule-associated protein 1 light chain 3 beta. By Figdraw.

**Figure 3 biomolecules-15-00263-f003:**
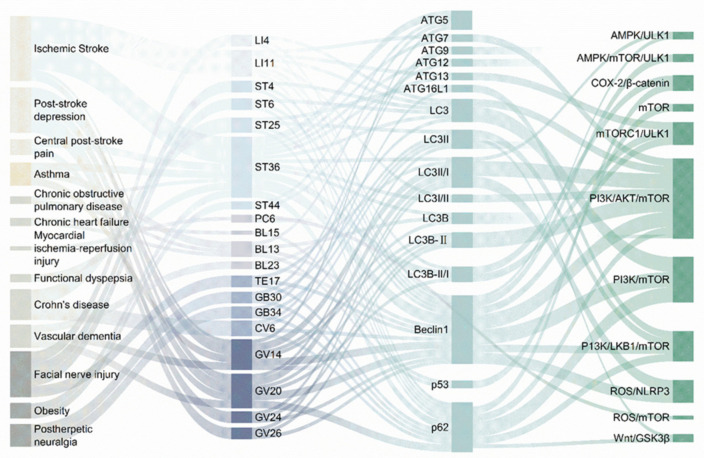
Disease acupoints signaling pathway linkages in the study of autophagy inhibition by acupuncture. The analytical form of Sankey diagrams is used to show the interactions and composition of acupoints, autophagy-related regulatory factors and signaling pathways in treating diseases by acupuncture inhibits autophagy. By Figdraw.

**Figure 4 biomolecules-15-00263-f004:**
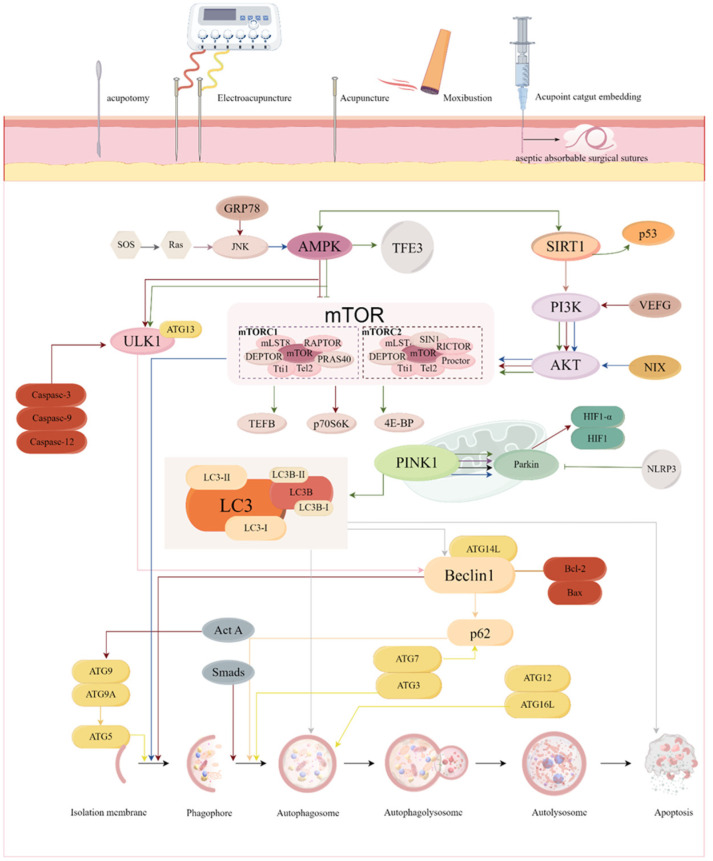
Signaling pathways associated with activation of autophagy by acupuncture. Acupuncture activates signaling pathways related to autophagy. There is increasing evidence that impaired autophagy leads to the accumulation of damaged proteins, organelles, and other pathological products, which induces or even exacerbates the onset and progression of diseases, such as Alzheimer’s disease (AD), polycystic ovary syndrome (PCOS), and rheumatoid arthritis (RA). Acupuncture can regulate autophagy-related signaling pathways to activate autophagy, accelerate the removal of pathological products and inflammatory factors, delay cellular aging, and reduce oxidative stress damage to maintain the stability of the intracellular environment. The different colors of the lines represent different acupuncture therapies: moxibustion: red; acupuncture: blue; electroacupuncture (EA): green; acupoint catgut embedding: purple; and acupotomy: black. For example, moxibustion activates autophagy via mTOR/p70S6K to improve cognitive impairment in AD rats. SOS, Son-of-Sevenless; Ras, Ras protein; JNK, c-Jun N-terminal kinase; AMPK, AMP-activated protein kinase; ULK1, Unc-51 Like Kinase 1; mTOR, mammalian target of rapamycin; NLRP3, NOD-like receptor thermal protein domain associated protein 3; AKT, threonine kinase; PI3K, phosphoinositide3-kinase; p62, Ubiquitin-binding protein p62; ATG, Anti-human T-lymphocyte Globulin; LC3, microtubule-associated protein 1 light chain 3; LC3B, microtubule-associated protein 1 light chain 3 beta; GRP78, Glucose Regulated Protein 78kD; TFE3, transcription factor binding to IGHM enhancer 3; SIRT1, silent mating type information regulation 2 homolog- 1; VEFG, tumor necrosis factor; NIX, NIP3-like protein X; caspase-3/9/12, cysteinyl aspartate specific proteinase-3/9/12; TEFB, transcription elongation factor b; p70S6K, p70 ribosomal protein S6 kinase; 4E-BP, eukaryotic translation initiation factor 4E-binding protein 1; PINK, PTEN induced putative kinase 1; Parkin, Parkin RBR E3 ubiquitin protein ligase; HIF1, hypoxia inducible factor-1; Act A, actin assembly inducing protein; Bcl-2, B-cell lymphoma-2; Bax, Bcl-2 associated X protein. By Figdraw.

**Figure 5 biomolecules-15-00263-f005:**
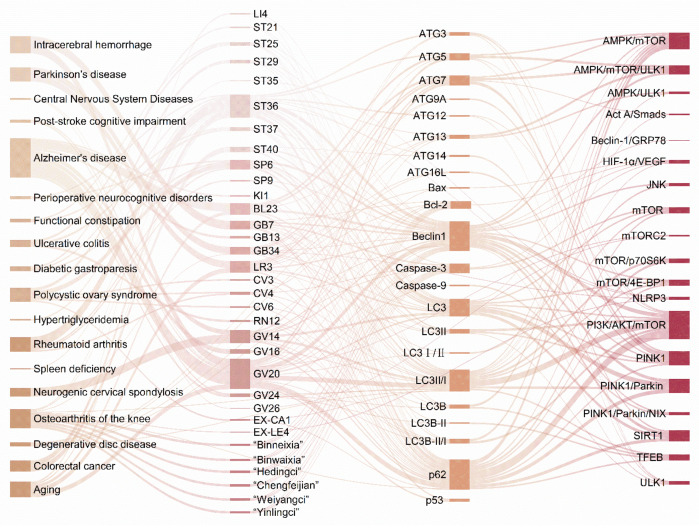
Disease acupoints signaling pathway linkages in the study of activation of autophagy by acupuncture. The analytical form of Sankey diagrams is used to show the interactions and composition of acupoints, autophagy-related regulatory factors, and signaling pathways in treating diseases by activation of autophagy by acupuncture. By Figdraw.

**Figure 6 biomolecules-15-00263-f006:**
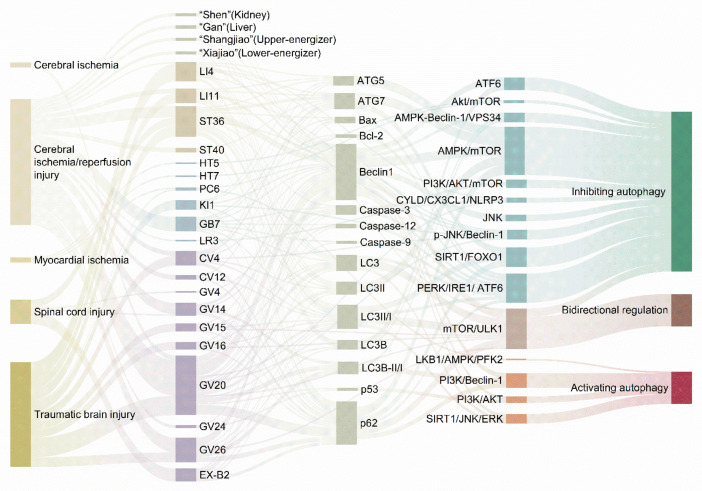
Disease-acupoints-signaling pathway linkages in the study of bidirectional regulation of autophagy by acupuncture. The analytical form of Sankey diagrams is used to show the interactions and composition between acupoints, autophagy-related regulatory factors, signaling pathways, and outcomes of modulation of autophagy in acupuncture modulation of autophagy treatment of disease. By Figdraw.

## Data Availability

No applicable.

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
