# Peer review of "The Mechanism of Acupuncture Regulating Autophagy: Progress and Prospect"

_biomolecules, 2025, doi:10.3390/biom15020263_

Round 1
Reviewer 1 Report
Comments and Suggestions for Authors
The manuscript by He and colleagues presents a review on the interaction between autophagy and cellular homeostasis/pathogenesis. Specifically, they summarized recent literature for an overview of the therapeutic mechanisms underlying acupuncture for positive, negative or bidirectional regulation via the autophagy pathway.
The strength of this work is for being comprehensive on the autophagy engaged cellular mechanisms. The major weakness includes (1) many mechanisms are related but the authors presented them in a segmented manner, (2) acupuncture’s anti-/pro-autophagy regulations (section 4) are presented rather tediously—Figure 3-6 as well as the paragraphs look repetitive in structure, (3) as a review there is insufficient novel point of view *from the authors* for the discussion and outlook.
Comments and suggestions in detail.
Line 76, “The aresulting degradation products…” “aresulting” should be “resulting”.
Line 133, “3.1.3 Hypoxic stress”, this subsection has an identical title to 3.1.1. Do the authors mean “osmotic stress”?
Line 149, “DNA damage occurs at an order of magnitude of 104-105 per cell per day”, “4” and “5” should be superscript.
Line 152, “ensuring the usual cellular fun”, “fun” should be “function”.
Line 166, the semicolons should be colons.
Line 180, several aspects of the stress induced autophagy are interconnected, such as 3.1.4 “DNA damage stress” and 3.1.6 “Oxidative stress”. They are better to be described together.
Line 196, the subsection 3.1.7 “Iron death” mechanisms are tightly connected to section 3.4 “Cell death”. Again, the authors are recommended to reorganize the structure to highlight the interrelationship.
Line 221, “However,…” Here the authors need to heed the logic—the two preceding examples are about autophagy’s pro-proliferation role, it’s hence odd to say “However, autophagy… can cause tumor growth”.
Line 250, “In contrast, …” In contrast to what?
Line 254-255, it’s better a brief introduction to ADCD versus AMCD can be provided here for smooth logical flow.
Line 280, the subsection 3.6 “Metabolism” is overly succinct compared to other cellular mechanisms. The authors are recommended to expand a bit, especially give some examples on key autophagy molecules involved in the “mutually regulated relationship” (Line 290).
Line 298-299, “…, the same acupoints have different regulatory effects on autophagy in different diseases.” Literature references are needed here for support.
Line 349, “post-stroke depression in rats by "Tongdu Zhongshan" acupuncture”. How to translate human acupoints to animal models needs some clarification.
Line 473, “PD is associated with abnormal autophagy in neurons.” Literature reference is needed here.
Line 548-550, “Acupuncture in the treatment of FC promotes intestinal motility, regulates intestinal flora, modulates the brain-gut axis…” Literature references are needed here.
Line 628, section 4.3, “Bidirectional regulation of autophagy by acupuncture”. In addition to inhibition (Figure 3) and activation (Figure 5), it is interesting to see the dual role of acupuncture in modulating autophagy. It would be better if the authors can make more summarized presentation (table or figure) for (1) which disorder(s) are likely due to excessive autophagy, which are likely due to lack of autophagy and which are likely due to both; (2) which type of acupuncture (EA, regular acupuncture, moxibustion, etc.) at which acupoint(s) are likely to inhibit autophagy, which type of acupuncture at which acupoint(s) are likely to promote autophagy, and which type of acupuncture at which acupoint(s) are likely to elicit dual-role effects.
Comments on the Quality of English LanguageSeveral issues are found in writing. The authors are recommended to double check the language and correct the mistakes.
Author Response
Comments 1: The strength of this work is for being comprehensive on the autophagy engaged cellular mechanisms. The major weakness includes:
(1) many mechanisms are related but the authors presented them in a segmented manner,
(2) acupuncture’s anti-/pro-autophagy regulations (section 4) are presented rather tediously—Figure 3-6 as well as the paragraphs look repetitive in structure,
(3) as a review there is insufficient novel point of view *from the authors* for the discussion and outlook.
Response 1:
We thank the reviewer for the encouragement and overall positive and constructive reviews. The comments raised have greatly improved the quality of the manuscript. We have revised the manuscript in accordance with the reviewer’s comments. And point-by-point responses to the comments were as follows:
(1) In response to the feedback received, the article's structure has been revised to enhance readability. Specifically, Sections 3.1.4, "DNA Damage Stress," and 3.1.6, "Oxidative Stress," have been combined into a single section in the original manuscript, see Revision 3.1.4, "DNA Damage and Oxidative Stress." Additionally, Section 3.1.7, "Iron Death," has been reorganized and relocated to Section 3.4, "Cell Death." While some textual content from the manuscript has been retained, necessary adjustments have been made in the corresponding sections to improve fluency and coherence. For a detailed description of these changes, please refer to Comments 7 and 8 below.
(2) After careful consideration, we have decided to retain the original structure of the article; however, the description of the corresponding section has been modified to enhance readability for our audience. Regarding this matter, we will explain as follows: The fourth subsection of the manuscript reviews existing research on acupuncture and moxibustion in relation to autophagy regulation. We categorize the effects of acupuncture on autophagy into three classifications: acupuncture inhibition of autophagy, acupuncture activation of autophagy, and acupuncture's bidirectional regulation of autophagy. Within each specific subsection, we systematically classify the diseases addressed in the studies. The purpose of structuring the article in this manner is twofold: firstly, to enable readers to intuitively and clearly grasp the current research landscape concerning acupuncture's regulation of autophagy; and secondly, to highlight the contrast between acupuncture's inhibition and activation of autophagy. Notably, research concerning acupuncture and moxibustion's activation of autophagy encompasses a broader range of systems, including the musculoskeletal system and tumors. This insight may prove beneficial for future research aimed at expanding the application of acupuncture in regulating autophagy for disease treatment.
(3) This paper systematically reviews the research advancements in the application of acupuncture for treating various diseases via the autophagy pathway. It summarizes the relevant signaling pathways through which acupuncture regulates autophagy and analyzes the shortcomings present in existing studies regarding the regulation of cellular autophagy by acupuncture. This analysis aims to provide a reference for further research on the modulation of autophagy through acupuncture. To enhance clarity for reviewers and readers, the authors outline the innovations of this paper as follows:
(1)This paper summarizes the research advancements in the application of acupuncture for treating various diseases via the autophagy pathway. Additionally, it analyzes the limitations present in the existing studies, thereby offering insights and experimental foundations for future research on the modulation of cellular autophagy through acupuncture.
(2)The findings indicate that acupuncture exerts a bidirectional modulating effect on autophagy capacity; it inhibits excessive autophagy while simultaneously activating defective autophagy. The direction of this modulation is contingent upon the organism's pathological state.
(3)A systematic research framework was developed to investigate the regulation of cellular autophagy by acupuncture. This study aims to enhance the precision of signaling pathways and the specificity of indicators by establishing more appropriate animal models, thereby elucidating the regulatory mechanisms of acupuncture at various stages of autophagy.
Comments 2: Line 76, "The aresulting degradation products…" "aresulting" should be "resulting".
Response 2: Agree. We sincerely apologize for the typo in the manuscript. The corresponding part has been revised, see line 83, page 2 of the text. We are very grateful for your careful review and for pointing out this mistake. We will ensure a more thorough check of the manuscript to avoid similar issues and further improve its quality.
“The resulting degradation products, including amino acids, proteins, carbohydrates, and other cellular constituents, are transported to the cytosol to synthesize new cellular constituents or provide energy.”
Comments 3: Line 133, "3.1.3 Hypoxic stress", this subsection has an identical title to 3.1.1. Do the authors mean "osmotic stress"?
Response 3: Agree. Title 3.1.3 should be Osmotic stress. The corresponding part has been revised, see line 138, page 4 of the text. We apologize for any writing errors in the manuscript. Thank you for your correction and we will check and proofread the contents of the manuscript again.
“3.1.3 Osmotic stress”
Comments 4: Line 149, "DNA damage occurs at an order of magnitude of 104-105 per cell per day", "4" and "5" should be superscript.
Response 4: Thank you for your careful review and we have responded to the issues you pointed out. The corresponding part has been revised, see line 155, page 4 of the text.
“Eukaryotic cells are exposed to various endogenous and exogenous stresses, and DNA damage occurs at an order of magnitude of 104-105 per cell per day [38].”
Comments 5: Line 152, "ensuring the usual cellular fun", "fun" should be "function".
Response 5: Thank you for your careful reading of the manuscript and we have been incorporated into the text on line 164, page 4 of the text.
“DNA damage response and repair (DDR/R) is a hierarchical structural mechanism evolved by the cell to recognize and repair DNA damage during the cell cycle, ensuring the usual cellular function [41]”
Comments 6: Line 166, the semicolons should be colons.
Response 6: Thank you for your careful reading and we have been corrected in the manuscript, see line 182-184, page 5 of the text. In addition, we have embellished this sentence.
“The ER is crucial for protein folding, translocation, and post-translational modifications, as well as for Ca2+ storage and lipid and carbohydrate metabolism, thereby playing an essential role in maintaining cellular homeostasis and function [48].”
Comments 7: Line 180, several aspects of the stress induced autophagy are interconnected, such as 3.1.4 "DNA damage stress" and 3.1.6 "Oxidative stress". They are better to be described together.
Response 7: We would like to sincerely thank you for your efforts in critically reviewing this manuscript and for providing valuable feedback. We have integrated the sections on DNA damage and oxidative stress, reorganized the language for improved clarity, and elucidated the connection between DNA injury and oxidative stress, as well as the relationship between autophagy and these two factors. The relevant section can be found on line 153-180, page 4-5 of the manuscript.
“3.1.4 DNA damage and oxidative stress
Eukaryotic cells are exposed to various endogenous and exogenous stresses, and DNA damage occurs at an order of magnitude of 104-105 per cell per day [38]. This damage negatively impacts cellular function and homeostasis. Reactive oxygen species (ROS), a collective term for oxidized metabolites and their derivatives [39], promotes the activity of transcription factors that play an active role in cell proliferation and differentiation. However, continuous stimulation from endogenous cell metabolism and/or exogenous environmental toxins can lead to the excessive accumulation of ROS, inducing oxidative stress that compromises the integrity and functions of cells, directly or indirectly results in DNA damage [40]. DNA damage response and repair (DDR/R) is a hierarchical structural mechanism evolved by the cell to recognize and repair DNA damage during the cell cycle, ensuring the usual cellular function [41]. In recent years, the interplay between DNA damage and oxidative stress has been increasingly recognized in a variety of diseases, including spinal cord injury (SCI) and traumatic brain injury (TBI) [42].
Autophagy is recognized as a crucial component of the integrated cellular stress response, providing adaptive and protective mechanisms against DNA damage and oxidative stress in cells. It can be adaptive and protective against cellular DNA damage and can act as an energy source during cell cycle arrest and repair mechanisms [43, 44]. DNA damage has been shown to activate autophagy through mTORC1 signaling, stimulate the expression of pro-autophagic p53-induced target genes, and enhance the interaction of the p105 subunit of NF-kappaB with autophagy proteins such as Beclin1[45, 46]. Furthermore, oxidative stress induced autophagy plays a vital role in preventing cellular damage and maintaining homeostasis in vivo [47]. ROS can induce the formation and expansion of autophagosomes, initiating autophagy to protect cells from oxidative stress [48]. Additionally, autophagy is believed to be involved in anti-oxidant functions and DDR/R. Genetic defects in autophagy genes can lead to tumor development, which is associated with ROS accumulation and subsequent DNA dam-age and organelles[49] .”
Comments 8: Line 196, the subsection 3.1.7 "Iron death" mechanisms are tightly connected to section 3.4 "Cell death". Again, the authors are recommended to reorganize the structure to highlight the interrelationship.
Response 8: Thank you for your careful and thorough reading of this manuscript, which has contributed to its overall quality. We have reorganized the structure by linking the concept of iron death to cell death and refined the language for clarity. In section 3.4 "Cell Death," we have supplemented the discussion on the relationship between iron death and cell death, which illustrates the close connection between autophagy and various types of cell death through specific examples. The corresponding part see line 229-264, page 6 of the text.
“3.4 Cell death
During the development and growth of an organism, cell death plays a crucial role in maintaining biological homeostasis by facilitating tissue remodeling and the removal of abnormal cells, thereby protecting the body from pathogenic substances. In the context of infection, chronic inflammation, and tissue injury, the sudden death of large cells or the accumulation of cell death can exacerbate disease progression [67]. Cell death patterns are categorized into accidental cell death (ACD) and regulated cell death (RCD). Unlike ACD, RCD is more critical for organismal development and tissue remodeling, encompassing three types of cell death: apoptosis, necroptosis, and pyroptosis. Additionally, ferroptosis is a distinct form of iron-dependent RCD that differs from apoptosis. The primary driver of this iron-induced cell death is the excessive ac-cumulation of intracellular bioactive iron-catalyzed lipid peroxides, leading to necro-sis-like morphological alterations in the affected cells [68]. In mammals, dysregulated ferroptosis significantly contributes to various pathological processes and conditions, including acute tissue damage, infections, and cancer.
Autophagy is typically associated with cell death and plays a crucial role in the regulation of virtually all modes of cell death within disease contexts. The involvement of autophagy in cell death can be divided into ADCD and autophagy-mediated cell death (AMCD). ADCD is significantly influenced by various components of the autophagy pathway. Autophagy flux is elevated during cell death and operates independently of other forms of programmed cell death. ER-phagy and mitophagy are classified as forms of ADCD. In contrast, AMCD positions autophagy as a foundation-al process that initiates other modes of cell death, interacts with cell death molecules, or directly leads to apoptosis, necrosis, and ferroptosis [69]. ATG12, an autophagy regulatory protein, promotes apoptosis and exhibits anti-cancer effects [70]. The mechanisms underlying necroptosis and apoptosis partially overlap, with autophagy mediating the transition between these two processes in specific cellular environments. Necrotic apoptosis plays a facilitating role in the early stages of autophagy while exhibiting an opposing effect in the later stages [71]. Autophagy regulates the caspa-se-1-mediated classical pyroptosis pathway through the NLRP3 inflammasome and ROS in the liver [72]. Recently, ferroptosis has been recognized as often being accompanied by the overactivation of autophagy and lysosomes [73]. An international consensus guideline has proposed the concept of autophagy-dependent ferroptosis, which offers a comprehensive overview of the complex relationship between autophagy and ferroptosis [74]. Autophagy inhibits lipid peroxidation, maintains cellular homeostasis, and selectively removes damaged or dysfunctional cellular components during ferroptosis.”
Comments 9: Line 221, "However, …" Here the authors need to heed the logic the two preceding examples are about autophagy's pro-proliferation role, it's hence odd to say "However, autophagy… can cause tumor growth".
Response 9: The manuscript has been revised in accordance with the reviewer’s comments. We reorganized language expression to make the logic of the article smoother. The corresponding part has been revised, see line 206-208, page 5 of the text.
“But on the other hand, autophagy is one of the factors driving tumor growth, potentially accelerating tumor progression and contributing to cancer recurrence as well as the emergence of drug resistance [60].”
Comments 10: Line 250, "In contrast, …" In contrast to what?
Response 10: In the manuscript, Line 250, the phrase "In contrast..." denotes a comparison between programmed cell death (RCD) and accidental cell death, highlighting the former's greater significance in organismal development or tissue remodeling. The reviewer's concerns are very reasonable. We reorganized language expression to make the logic of the article smoother. The corresponding part has been revised, see line 235-236, page 6 of the text.
“Unlike ACD, RCD is more critical for organismal development and tissue remodeling, …”
Comments 11: Line 254-255, it's better a brief introduction to ADCD versus AMCD can be provided here for smooth logical flow.
Response 11: Response: Thank you for your valuable feedback. We have incorporated the comment into the revised manuscript, The corresponding part has been revised, see line 245-252, page 6 of the text.
“The involvement of autophagy in cell death can be divided into ADCD and autophagy-mediated cell death (AMCD). ADCD is significantly influenced by various components of the autophagy pathway. Autophagy flux is elevated during cell death and operates independently of other forms of programmed cell death. ER-phagy and mitophagy are classified as forms of ADCD. In contrast, AMCD positions autophagy as a foundational process that initiates other modes of cell death, interacts with cell death molecules, or directly leads to apoptosis, necrosis, and ferroptosis [69].”
Comments 12: Line 280, the subsection 3.6 "Metabolism" is overly succinct compared to other cellular mechanisms. The authors are recommended to expand a bit, especially give some examples on key autophagy molecules involved in the "mutually regulated relationship" (Line 290).
Response 12: The reviewer's proposal is insightful. We have improved this section on metabolism by thoroughly reviewing relevant literature. We employed tumors as examples to illustrate that the innovative strategy of modulating metabolism and targeting autophagy presents a novel avenue for anti-cancer therapy. The corresponding part has been revised, see line 293-297, page 7 of the text.
“Disruption of lipid metabolism in cancer cells can lead to systemic energy dysregulation. Stearoyl-CoA desaturase 1 (SCD1) is crucial for maintaining cellular lipid homeostasis and is closely linked to autophagy. This interaction serves as a basis for exploring innovative combinatorial anticancer strategies that utilize SCD1 inhibitors along-side autophagy modulators [84].”
Comments 13: Line 298-299, "…, the same acupoints have different regulatory effects on autophagy in different diseases." Literature references are needed here for support.
Response 13: The revisions proposed by the reviewer have been incorporated into the text on line 304-305, page 7 of the text. These two documents use Zusanli (ST36) as an example to demonstrate that the same acupoint can have different regulatory effects on autophagy in various diseases. Specifically, ST36 promotes hepatic autophagy in rats with hyperlipidemia, while it suppresses excessive autophagy in rats with functional dyspepsia.
“However, the same acupoints have different regulatory effects on autophagy in different diseases [85, 86].”
- Dong, J.; Zhang, Y.; Wei, Y.; Xu, H.; Liu, L.; Deng, T.; Zhang, L. Effects of electroacupuncture at "Zusanli" (ST 36) on expression of mitophagy-related proteins in skeletal muscle in rats with spleen deficiency syndrome. Zhongguo Zhen Jiu 2018, 38, 741-746.
- Pan, X.L.; Zhou, L.; Wang, D.; Han, Y.L.; Wang, J.Y.; Xu, P.D.; Zhang, H.X.; Zhou, L. Electroacupuncture at "Zusanli"(ST36) promotes gastrointestinal motility possibly by suppressing excessive autophagy via AMPK/ULK1 signaling in rats with functional dyspepsia. Zhen Ci Yan Jiu 2019, 44, 486-491.
Comments 14: Line 349, "post-stroke depression in rats by "Tongdu Zhongshan" acupuncture". How to translate human acupoints to animal models needs some clarification.
Response 14: We attach great importance to the queries raised by the reviewers. The concept of “Tongdu Tiaoshen” acupuncture is rooted in traditional Chinese medicine and is based on the academic insights of Prof. Zhang Daozong, a renowned veteran practitioner in this field. He posits that 'the lesion is in the brain, and the first to address it is through the Governor's pulse. The acupoints are selected from four locations along the Governor's pulse: "Dazhui" (GV 14), "Shuigou" (GV 26), "Baihui" (GV 20), and "Shenting" (GV 24). The example here is to illustrate that electroacupuncture inhibits neuronal autophagy. For a more intuitive explanation and easier understanding for readers, we have modified it in the manuscript. The corresponding part has been revised, see lines 349–350, page 12 of the text.
“Acupuncture was applied at "Dazhui" (GV 14), "Shuigou" (GV 26), "Baihui" (GV 20), and "Shenting" (GV 24), demonstrating an effect on post-stroke depression in rats, ”
Comments 15: Line 473, "PD is associated with abnormal autophagy in neurons." Literature reference is needed here.
Response 15: Thank you very much for your valuable comments. Your Suggestions have helped us to improve the deficiencies in the manuscript. The corresponding part has been revised, see line 489-490, page 22 of the text.
“PD is associated with abnormal autophagy in neurons [114].”
- Hsu, W.T.; Chen, Y.H.; Yang, H.B.; Lin, J.G.; Hung, S.Y. Electroacupuncture Improves Motor Symptoms of Parkinson's Disease and Promotes Neuronal Autophagy Activity in Mouse Brain. AM J CHINESE MED 2020, 48, 1651-1669.
Comments 16: Line 548-550, "Acupuncture in the treatment of FC promotes intestinal motility, regulates intestinal flora, modulates the brain-gut axis…" Literature references are needed here.
Response 16: Thank you for your careful review and constructive suggestions regarding our manuscript. The corresponding part has been revised, see line 564-567, page 23 of the text.
“Acupuncture in the treatment of FC promotes intestinal motility, regulates intestinal flora, modulates the brain-gut axis, attenuates intestinal inflammatory responses, and improves rectal hypersensitivity [154].”
- Ong, S.S.; Tang, T.; Xu, L.; Xu, C.; Li, Q.; Deng, X.; Shen, P.; Chen, Y.; Song, Y.; Lu, H.; Fang, L. Research on the mechanism of core acupoints in electroacupuncture for functional constipation based on data mining and network acupuncture. FRONT MED-LAUSANNE 2024, 11, 1482066.
Comments 17: Line 628, section 4.3, "Bidirectional regulation of autophagy by acupuncture". In addition to inhibition (Figure 3) and activation (Figure 5), it is interesting to see the dual role of acupuncture in modulating autophagy. It would be better if the authors can make more summarized presentation (table or figure) for (1) which disorder(s) are likely due to excessive autophagy, which are likely due to lack of autophagy and which are likely due to both; (2) which type of acupuncture (EA, regular acupuncture, moxibustion, etc.) at which acupoint(s) are likely to inhibit autophagy, which type of acupuncture at which acupoint(s) are likely to promote autophagy, and which type of acupuncture at which acupoint(s) are likely to elicit dual-role effects.
Response 17: Thank you for careful and thorough reading of this manuscript, which help to improve the quality of this manuscript. We changed the "Bidirectional regulation of autophagy by acupuncture" section according to the reviewer’s suggestion and see line 654-670, 673-715, page 25-26 and page 30 of the text.
“4.3 Bidirectional regulation of autophagy by acupuncture
Autophagy, whether deficient or excessive, disrupts intracellular environmental homeostasis and plays a dual role in cytoprotection and cell death. The mechanisms by which autophagy contributes to the onset and progression of diseases remain a topic of considerable interest. Autophagy excessive can lead to cell dysfunction and cell death, contributing to the development of various diseases as the excessive accumulation of autophagosomes in cardiomyocytes contributes to heart failure [161], while a similar accumulation in the neuronal cytoplasm exacerbates the symptoms associated with neurological deficits in cases of ischemic brain injury [87]. Conversely, autophagy deficiency hinders the degradation and digestion of long-lived proteins and damaged organelles, which contributes to the development of several diseases, with AD being the most representative. Furthermore, autophagy deficiencies exacerbate amyloidosis and tau pathology [121]. Interestingly, the development of certain diseases is inextricably linked to both overactivation and underactivation of autophagy. For example, the role of autophagy in traumatic brain injury (TBI) remains controversial [162, 163]. A review of the relevant literature suggests that the modulation of autophagy through acupuncture can have opposing effects within the same disease context, as observed in conditions such as cerebral ischemia/reperfusion, myocardial ischemia, and TBI (Table 3).
EA regulates autophagy in the context of cerebral ischemia-reperfusion injury (CIRI), resulting in two primary outcomes. First, it modulates apoptosis following CIRI by activating autophagy. By stimulating the PI3K/Akt and type III PI3K/Beclin1 signaling pathways, EA enhances autophagy, reduces the number of apoptotic cells, and de-creases infarct volume, thereby exerting a neuroprotective effect [169, 170]. Second, EA inhibits excessive neuronal autophagy by reducing the quantity of autophagosomes and autophagolysosomes, regulating the expression levels of autophagy-related factors and microRNAs, and mitigating brain injury along with the associated inflammation [155, 165, 171, 180-182]. In contrast, only moxibustion pretreatment has been re-ported to improve CIRI, and its mechanism is associated with autophagy inhibition [88]. The acupoints selected for EA and moxibustion treatment of CIRI predominantly include the Governor's pulse, with GV14 and GV20 being the most frequently targeted. Autophagy has been shown to protect cardiomyocytes during the early stages of acute myocardial ischemia (AMI). However, in the later stages, excessive autophagic activity may exacerbate myocardial necrosis and dysfunction. The regulation of cardiomyocyte autophagy by acupuncture is also bidirectional. EA intervention at HT5 and HT7 in the heart meridian has demonstrated therapeutic effects by activating the AKT/mTOR signaling pathway and inhibiting autophagy levels in cardiomyocytes of AMI rats [ 183]. Conversely, the effects of PC6, which interacts with the pericardium meridian, in reducing apoptosis and necrosis while promoting the survival of damaged cardiomyocytes, are linked to the activation of autophagy through the LKB1/AMPK/PFK2 signaling pathway [184].
Apoptosis and autophagy activities in the SCI lesion area were inhibited, which limited nerve regeneration and functional recovery [177]. EA at the "Jiaji" acupoints inversely activated autophagy and regulated the disordered microenvironment of the SCI, thereby promoting the recovery of motor function following SCI [176]. Further-more, EA at the GV4 and GV14 acupoints inhibited autophagy by enhancing the activity of the PI3K/AKT/mTOR signaling pathway, which facilitated the recovery of hindlimb motor function and spinal cord neurological function in rats with SCI. This effect may be associated with the regulation of cell proliferation and differentiation, as well as the promotion of axon regeneration [185]. In post TBI, acupuncture can either promote or inhibit autophagy, leading to improvements in neurological function, cognitive abilities, and behavioral outcomes in animal models of TBI. EA facilitates the recovery of neurological function and mitigates pathological brain tissue damage and neuronal apoptosis in TBI rats by inhibiting autophagy [186, 187]. However, acupuncture exerts a time-dependent effect on neuronal autophagy in the damaged cerebral cortex following TBI. This is evidenced by observations that acupuncture enhances autophagic flux and induces autophagic activation on day 3 post-injury, while preventing excessive autophagy on days 7 and 14 post-injury [179]. Thus, acupuncture appears to enhance autophagy and improve TBI prognosis when autophagy levels are insufficient while inhibiting autophagy to exert therapeutic effects when autophagy is excessive. Collectively, these findings indicate that the role of autophagy in disease progression can vary at different pathological stages and that the modulation of autophagy by acupuncture may be closely linked to the timing of the intervention.”

Reviewer 2 Report
Comments and Suggestions for Authors
This is a very comprehensive review which starts off with a detailed description of the benefits and hazards of autophagy in various disease states. It then introduces the modulation of autophagic events by use of acupuncture.
1. The article is clearly written but some of the figures do not add much to the text as they are too complex to follow. This is especially true of figures 5 and 6.
2. One topic that needs to be addressed is why acupuncture always proceeds in a beneficial direction whether by inhibiting or stimulating autophagy. This is not easy to explain but important in order to validate the entire acupuncture approach to medicine.
3. An article as long and thorough as this would benefit by the inclusion of an index at the beginning.
4. All of the Chinese language articles cited in Zhen Ci Yan Jiu and Zhongguo Zhen Jiu are in PubMed and the abstracts are available in English. This is good but it would be nice for readers to see English versions of the entire text.
Author Response
Dear Reviewer 2, Thank you so much for handling the review of our manuscript. We appreciate the promoting comments to our study, and we have accepted and revised as recommended in this revised manuscript. In this revised version, changes to our manuscript within the document were all highlighted by using red colored text. Point-by-point responses to the reviewers are listed below this letter. Please see the attachment.

Reviewer 3 Report
Comments and Suggestions for Authors
This review article collects a large number of literatures and reviews, and discusses the importance of autophagy and acupuncture in many diseases. However, the scope of the subject matter is too broad, and there are many superficial descriptions, so it was not possible to obtain insightful messages. In addition, I found many incorrect words and sentences in the first two pages (as listed below), and I got the impression that there would be many inappropriate descriptions in the rest of the text as well. Therefore, it seemed that this manuscript was far from completion. More detailed research, analysis, and planning are needed.
Minor comments (only some)
Line 59: There is established words, “phagophore” or “isolation membrane”, for the phagocytic vesicle.
Line 61: “Septum” is not appropriate.
Line 66-67: The sentence is totally wrong.
Line 69: Use “conjugation” rather than “coupling”.
Line 73-74: use “autolysosome” rather than “autophagic lysosome”.
Line 196-210: This section should be an introduction of “ferroptosis”.
Author Response
Dear Reviewer 3, Thank you for reviewing our manuscript and for the constructive comments, which greatly helped us to improve the manuscript. The manuscript was carefully revised, and point-by-point response was listed below. We hope that your comments have been addressed accurately. The revised manuscript was marked with red color, and the responses we represented in red text. Please see the attachment.

Round 2
Reviewer 1 Report
Comments and Suggestions for Authors
The reviewers have revised the manuscript. I don't have more questions now.
Author Response
Dear Reviewer 1,
We sincerely appreciate your thorough review of our revised manuscript and your decision not to provide further comments. Your professional insights and valuable suggestions have significantly contributed to enhancing the quality of our paper. Once again, we thank you for your patience and expert guidance.
Thank you and best regards.
Yours sincerely,
Corresponding author:
Tie Li
Tel: +8613019223202
E-mail: litie@ccucm.edu.cn
